# Characteristics and Antibiotic Treatment of Patients with Cellulitis in the Emergency Department

**DOI:** 10.3390/antibiotics13111021

**Published:** 2024-10-30

**Authors:** Aaron K. Wiederhold, Mariana B. Cartuliares, Karin Jeppesen, Helene Skjøt-Arkil

**Affiliations:** 1Faculty of Health Sciences, University of Southern Denmark, 5230 Odense, Denmark; 2Department of Emergency Medicine, University Hospital of Southern Denmark, 6200 Aabenraa, Denmark; 3Department of Regional Health Research, University of Southern Denmark, 6200 Aabenraa, Denmark; 4Department of Otorhinolaryngology, Head and Neck Surgery, University Hospital of Southern Denmark, 6400 Sønderborg, Denmark

**Keywords:** antibiotics, cellulitis, skin and soft tissue infection, emergency department

## Abstract

Background: Cellulitis is a common infection in Emergency Departments (EDs), and unclear diagnostics may lead to unnecessary treatment with broad-spectrum antibiotics. The aim of this study was to characterize patients with cellulitis admitted to the ED, describe the type and route of antibiotic treatment and compare the prognosis for cellulitis to that for other infections. Methods: This multicenter, cross-sectional study prospectively included adult patients admitted to the ED suspected of having an infection. Data were collected from medical records, and an expert panel assigned a final diagnosis to each patient. Only patients diagnosed with an infection were included in this study. Results: A total of 777 patients were included, of which 77 were diagnosed with cellulitis. Diabetes, obesity and prior cellulitis were associated with cellulitis with an OR of 2.01 [95% CI: 1.17–3.46], OR 2.66 [95% CI: 1.54–4.59] and OR 14.88 [95% CI: 7.88–28.08], respectively. The majority of the patients were treated, according to the regional guidelines, with narrow-spectrum antibiotics. Broad-spectrum antibiotics were rarely used. The patients with cellulitis were hospitalized for a shorter time compared to those with other infections, but 26% were readmitted within 30 days. The 30-day mortality of the patients with cellulitis was 3.9% and did not differ significantly from other infections.

## 1. Introduction

Cellulitis is one of the most common infections in Emergency Departments (EDs) and is a bacterial skin infection characterized by a sudden onset of diffuse redness, swelling and pain in a spreading area [1,2,3]. The infection typically affects the lower limb and is usually caused by beta-hemolytic streptococci [2,4]. Although cellulitis and erysipelas are often described as the same condition, erysipelas is characterized by only involving the outer layer of the skin and affecting a sharply demarcated area [2,5]. In this study, erysipelas is not defined as a separate diagnosis but is considered a type of cellulitis.

The known predisposing factors for cellulitis include disruptions of the skin barrier, such as wounds and leg ulcers, excoriating skin diseases, lymphoedema, obesity, diabetes and prior cases of cellulitis [6,7].

The majority of patients diagnosed with cellulitis are treated in primary care with oral antibiotics, but severe cases with systemic symptoms may need intravenous (i.v.) treatment and hospital admission [4,8]. Cellulitis can be challenging for the receiving physician to diagnose, as the diagnosis is based on the patient’s clinical presentation, and unclear skin infections can lead to inappropriate treatment with broad-spectrum antibiotics [4,9]. Clear knowledge of the characteristics of patients with cellulitis could help the physician decide on precise antibiotic treatment when receiving a patient suspected of having an infection in the ED. Although Denmark has some of the lowest prevalences of antimicrobial resistance, the numbers are increasing [10,11]. To avoid further development of resistance, adherence to strict national guidelines for antibiotic treatment issued by the Danish Health Authority, which aims to reduce broad-spectrum antibiotics, is required [12]. In Denmark, the antibiotic guidelines for the treatment of cellulitis prescribe narrow-spectrum antibiotics in terms of monotherapy with penicillin. Guidelines for other European countries recommend a broader antibiotic treatment with amoxicillin, and guideline adherence has been associated with better patient outcomes including a reduced need for surgical interventions [13]. Accurate preliminary diagnosis in the ED is essential for targeted treatment [14]. Different treatment strategies for cellulitis across countries and the lack of consensus on appropriate management highlight the necessity of further research, which this study aims to contribute to [3]. Furthermore, more knowledge is needed about the adherence to and outcomes of patients treated according to restrictive guidelines.

The objectives of this study were as follows:(1)To characterize patients with cellulitis at admission to the ED and compare them to patients diagnosed with other infections.(2)To describe the prescribed antibiotic treatment to patients with cellulitis admitted to the ED at 4 h, 48 h and 5 days after admission. In addition, we wanted to describe the amount of supplemental antibiotic treatment prescribed in addition to the standard therapy.(3)To investigate the length of hospital stay, discharge to other hospital wards, 30-day readmission rates and 30-day mortality of patients with cellulitis and those with other infections admitted to the ED.

## 2. Results

A total of 966 patients suspected of having an infection were enrolled in the INDEED study. We excluded 12 patients who tested positive for SARS-CoV-2 at admission. The expert panel diagnosed 177 (18%) patients with conditions other than infectious diseases, leaving 777 (80%) participants to be considered in the study analysis. Of these patients, 77 (10%) were diagnosed with cellulitis.

A flowchart of the recruitment, inclusion and exclusion of the participants is illustrated in Figure 1.

### 2.1. Patient Characteristics

The characteristics of the patients with cellulitis and other infections, along with their respective missing data, are shown in Table 1.

The proportion of male patients was the largest amongst the patients with cellulitis, whereas sex seemed to be more equally distributed amongst the patients with other infections. The median age of the patients with cellulitis was 66 (54–79) years, slightly younger than patients with other infections.

The vital parameters of the patients with cellulitis were found to be less affected in terms of fever and respiratory parameters such as oxygen saturation and respiration rate. Heart rate and systolic blood pressure did not differ between the groups. None of the patients with cellulitis had a qSOFA score ≥ 2.

Regarding lifestyle factors, fewer patients with cellulitis were current smokers and a larger proportion of the patients with cellulitis had a BMI above 30 compared to the patients with other infections. Type 2 diabetes was also found to be more frequent among the patients with cellulitis. Prior cases of cellulitis were highly associated with the patients admitted with cellulitis, OR 14.88 [95% CI: 7.88–28.08] *p* < 0.001.

No differences were found between the groups regarding inflammatory serum biomarkers.

### 2.2. Type and Route of Antibiotic Administration

The results of the antibiotic treatment for the patients with cellulitis are shown in Table 2. At the time of admission, 42 (54.5%) patients had received peroral antibiotic treatment within the last 30 days, mostly phenoxymethylpenicillin and dicloxacillin. At 4 h after admission to the ED, i.v. antibiotic treatment had been initiated for 61 (82.4%) of the patients, and approximately half of the patients receiving i.v. benzylpenicillin were being supplemented with i.v. cloxacillin. The administration of peroral and i.v. antibiotic treatment was equally distributed at 48 h after admission. After 5 days, the antibiotic treatment had been changed for the majority of the patients to consist of the peroral administration of phenoxymethylpenicillin and dicloxacillin, and 16 (21.1%) patients remained on i.v. antibiotics with mainly benzylpenicillin and cloxacillin.

A graphic representation of the administration of oral and i.v. antibiotics at four time points (prior to admission, 4 h, 48 h and 5 days) is illustrated in Figure 2.

### 2.3. Clinical Outcome

The patients with cellulitis had a significantly higher likelihood of discharge to home within 48 h compared to those with other infections, with an OR of 1.98 [95% CI: 1.23–3.18]. The results are shown in Table 3.

The majority of the patients, who still required hospitalization after 48 h, were transferred to medical wards for further treatment. None of the patients diagnosed with cellulitis had to be treated in the intensive care unit.

The readmission rate for the patients with cellulitis within 30 days after discharge was 26%. This number seemed to be slightly higher compared to the patients with other infections, but no significant difference was found between the two groups. No association was found between 30-day mortality and the patients diagnosed with cellulitis.

The length of hospital stay for the patients with cellulitis was found to be 18% lower, resulting in one less day of hospitalization compared to those with another infectious diagnosis. However, no statistical significance was found.

## 3. Discussion

The patients with cellulitis were slightly younger and systemically less affected at the time of admission compared to the patients admitted with other infections, and none of the patients with cellulitis scored ≥2 on the qSOFA score. Comorbidities in terms of type 2 diabetes, obesity and prior cases of cellulitis were highly associated with cellulitis.

Overall, the regional guidelines for the antibiotic treatment of patients admitted to the ED with cellulitis were, to a great extent, followed correctly. Nine out of ten patients were treated according to the guidelines, and only a few patients received broad-spectrum antibiotics throughout their admission. The majority of the patients received i.v. antibiotic treatment at the beginning of hospitalization and gradually shifted to peroral treatment within 5 days.

Half of the patients with cellulitis were discharged from the hospital within 48 h. The remaining patients were almost entirely transferred to medical wards for further treatment. One out of four patients with cellulitis was readmitted to the hospital within 30 days after discharge.

The distribution of the age and sex of the patients with cellulitis, with an overrepresentation of male patients in this study, corresponds to the existing literature [6,8,15], although a study from 2010 found a larger proportion of female patients [16]. The associated predisposing factors we found for cellulitis in terms of diabetes, obesity and prior cases of cellulitis also agreed with previous studies [6,7].

An explanation for the finding that patients with cellulitis tended to be less systemically affected could be the localized infection of the skin compared to patients with other infections, which for a large part consisted of respiratory infections and urinary tract infections.

Roughly 68% of the peroral administrations of phenoxymethylpenicillin in the ED were supplemented with peroral dicloxacillin. Amongst the patients receiving i.v. benzylpenicillin, 52% were supplemented with i.v. cloxacillin. While beta-hemolytic streptococci are known to be the main causatives for cellulitis, previous studies on the etiology of cellulitis found a large proportion of *Staphylococcus aureus* in swab cultures [4,15]. This would justify the frequent use of supplemental beta-lactamase-resistant penicillins in the included EDs. However, it is unclear which considerations lead to the decision of supplementing the standard treatment, as wound and skin swab cultures are not usual practice in diagnosing cellulitis in the included EDs.

The patients with cellulitis in this study were mostly only hospitalized for a few days. Although the number of discharges within 48 h suggests efficient treatment with i.v. antibiotics, 26% of the patients were readmitted to the hospital within 30 days. This may indicate that treatment with i.v. antibiotics for a slightly longer time could possibly reduce the rate of rehospitalization. A study from 2020 [17] tried to investigate the best timing for switching from i.v. to oral antibiotic treatment in patients with cellulitis. The meta-analysis, however, only found a single trial for this topic, which was too small to demonstrate any conclusive findings. Further investigation is needed to help the physician decide the timing of switching from i.v. to oral antibiotic treatment.

The 30-day mortality of the patients with cellulitis was found to be 3.9%. Interestingly, we found no significant association of lower mortality compared to the patients with other infections, even though the patients with cellulitis in this study were systemically less affected, and mortality for cellulitis was found to be generally low in other studies (1.1–2.5%) [16,18].

More severe cases of cellulitis with a need for i.v. antibiotic treatment are mainly admitted to the ED, whereas the majority of cellulitis cases are milder and can be treated in primary care facilities, which were not considered in this study [8]. The results of this study are, therefore, most likely to only be applicable to patients with more severe cases of cellulitis with the need for i.v.-administered antibiotic treatment. The qSOFA score is used to early identify patients with infection with a greater risk of poor outcomes in terms of mortality [19]. Interestingly, none of the patients with cellulitis scored ≥2, indicating the less affected systemic parameters. Acutely ill patients with the need for immediate and life-saving treatment were excluded from this study, and our findings for the low qSOFA scores, shorter hospital stays and no transfers to the intensive care unit for patients with cellulitis may be affected by selection bias.

Even though the majority of the patients with cellulitis were treated with narrow-spectrum antibiotics as the guidelines prescribed, this study showed favorable outcomes in terms of a short hospital stay. None of the patients required transfer to the intensive care unit. Clinicians should, therefore, continue administering narrow-spectrum antibiotics for patients they diagnose with cellulitis.

### Strengths and Limitations

The strengths of this pragmatic study with prospective data collection are the large number of patients included and the multicenter participation of four different EDs. Since the participants were recruited from hospitals in the same region of Denmark, they followed the same regional guidelines for the antibiotic treatment of cellulitis. Therefore, these results might be generalizable to other EDs with similar organization and guidelines.

However, this study also has limitations. The data variables regarding lifestyle factors were found to be missing for a large proportion of the patients, especially for BMI with almost 26% missing data. This may result in selection bias and could have a significant impact on the analysis and limit the generalizability of our findings [20].

In this study, we only considered patients with a final diagnosis of infectious disease, and our analyses of associations between cellulitis and several factors are thereby based on data excluding patients with other conditions that are also treated in the ED.

## 4. Materials and Methods

### 4.1. Study Design and Setting

This cross-sectional study is based on data prospectively collected in the INDEED study (Infectious Diseases in Emergency Departments), a multifaceted, multicenter study that recruited patients admitted to the ED of three major hospitals in the Region of Southern Denmark between 1 March 2021 and 28 February 2022. The hospitals recruiting patients were Odense University Hospital in Odense, Lillebælt Hospital in Kolding and Hospital Sønderjylland in Aabenraa and Sønderborg. The protocol of the INDEED study has been previously published [21]. Project assistants with healthcare education recruited the participants and collected data from the patient’s electronic journals and by patient interviews.

### 4.2. Participants

Adult patients aged ≥18 years admitted to the ED and suspected by the receiving physician to have an infection were invited to participate in the project. Only the patients able to give informed consent were included.

Patients were excluded if the attending physician considered that participation could delay life-saving treatment or if the patient was directly transferred to the intensive care unit. Patients with either severe immunodeficiencies, a Severe Acute Respiratory Syndrome Coronavirus 2 (SARS-CoV-2)-positive test within 14 days of admission or hospitalization within the last 14 days were also excluded. A complete overview of the exclusion criteria is listed in the study protocol [21].

Only the patients with a final diagnosis based on an infectious disease were included in this study, thereby excluding patients with other conditions.

### 4.3. Data Source and Variables

Patient data including age, sex, vital parameters and blood analyses were extracted from the patient’s electronic medical records. Data on underlying diseases in terms of type 2 diabetes and cardiovascular diseases, prior cases of cellulitis and lifestyle factors such as alcohol consumption, smoking status and BMI were collected through patient interviews. The cut-off values for blood sample analyses and vital parameters were chosen based on what is applied in Danish hospitals. Cardiovascular diseases included ischaemic heart disease and cardiac heart failure. Alcohol overuse was registered if the consumption exceeded the Danish alcohol recommendations at the time of data collection [22].

The Quick Sepsis-related Organ Failure Assessment (qSOFA) score was calculated based on the collected data and 1 point was given based on the following parameters: systolic blood pressure ≤100 mmHg, respiration frequency ≥22/min and Glasgow Coma Scale (GCS) <15 [19].

The type and route of administration of antibiotic treatment were registered at four time points: within the last 30 days before admission, and 4 h, 48 h and 5 days after admission. Antibiotic prescriptions were recorded and categorized into three groups:(1)Narrow-spectrum beta-lactamase-sensitive penicillins (Anatomical Therapeutic Chemical code (ATC) J01CE (phenoxymethylpenicillin for peroral use and benzylpenicillin for intravenous use));(2)Narrow-spectrum beta-lactamase-resistant penicillins (ATC J01CF (dicloxacillin for peroral use and cloxacillin for intravenous use));(3)Other (ATC J01CR05 (piperacillin/tazobactam), ATC J01DB (cephalosporins), ATC J01CA (pivampicillin and ampicillin), ATC J01X (metronidazole), ATC J01F (lincosamides and macrolides), ATC J01E (sulfonamides) and ATC J01G (aminoglycosides)).

Furthermore, data from the patient’s medical records on the patient’s length of hospital stay, discharge placement from the ED after 48 h, readmission within the first 30 days after discharge and 30-day mortality were also collected.

Samples for microbiological analysis were not obtained systematically from patients with cellulitis, and the results of microbiological samples have, therefore, not been taken into consideration in this study.

### 4.4. Outcome

A panel of experts consisting of two consultants per hospital with experience in emergency medicine and infectious diseases retrospectively assigned a final diagnosis for all the patients included in the study. The diagnosis was based on all the relevant information from the medical records, clinical examinations and paraclinical analyses at admission.

### 4.5. Antibiotic Guidelines

The regional guideline for the antibiotic treatment of hospitalized patients with cellulitis in the Region of Southern Denmark is shown in Figure 3.

### 4.6. Statistical Methods

Data on categorical binary variables were presented as numbers (*n*) and percentages (%). Continuous variables were tested for normal distribution and summarized in median and interquartile range (IQR) if a non-normal distribution was found. Univariate logistic regression was performed to identify the factors associated with cellulitis, and the results were presented as odds ratios (OR) with 95% confidence intervals (CI) and *p*-values. Negative binomial regression was used to calculate the incidence rate ratio (IRR) and presented with 95% CI and *p*-values. All the statistical analyses were performed using Stata (StataCorp. 2023. Stata/BE 18.0. College Station, TX, USA).

### 4.7. Ethics

The INDEED project was approved by the Regional Committees on Health Research Ethics for Southern Denmark (S-20200188), and the processing of personal data was notified to and approved by the Region of Southern Denmark and listed in the internal record (no. 20/60508) cf. Art 30 of The EU General Data Protection Regulation.

Informed consent was secured from each patient prior to participation in the study and after information about the project had been given orally and in writing.

## 5. Conclusions

We found that diabetes, obesity and prior cases of cellulitis were highly associated with cellulitis at the ED.

The regional guidelines for the antibiotic treatment of cellulitis were followed to a large extent, and broad-spectrum antibiotics were rarely used. For half of the patients, supplemental treatment was added to the standard treatment. The initial treatment was administered intravenously to four out of five patients.

Half of the patients with cellulitis were discharged to home within two days. There were no differences in the mortality, length of hospital stay or readmission compared to patients with other infections.

## Figures and Tables

**Figure 1 antibiotics-13-01021-f001:**
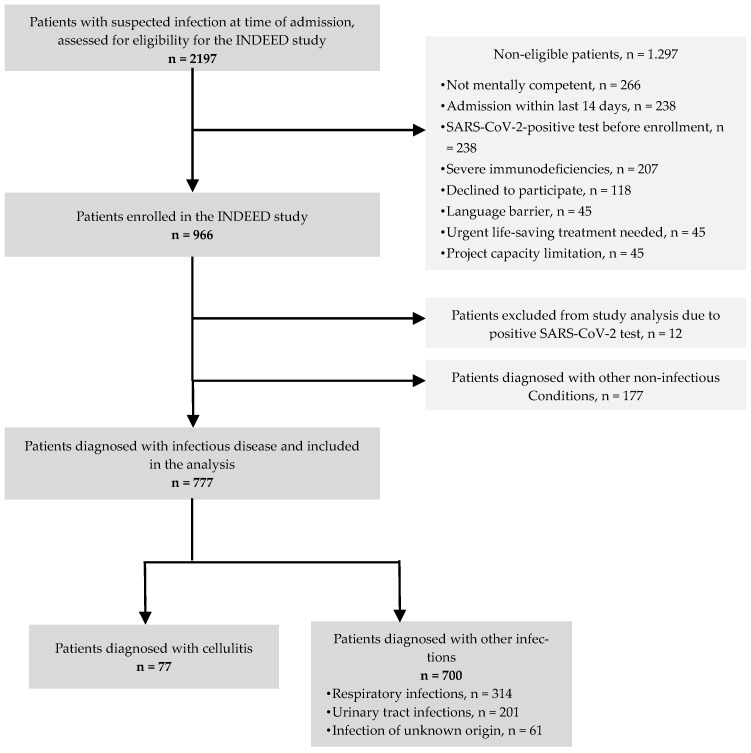
Study population.

**Figure 2 antibiotics-13-01021-f002:**
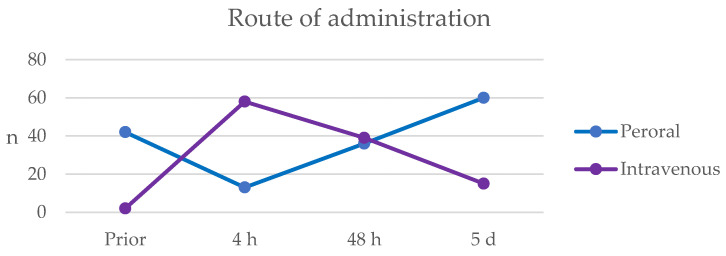
Route of administration of antibiotic treatment for the 77 patients diagnosed with cellulitis prior to admission and at three different time points: 4 h, 48 h and 5 days after admission. The blue line represents the number of patients treated with peroral antibiotics, and the purple shows the patients treated intravenously with antibiotics.

**Figure 3 antibiotics-13-01021-f003:**
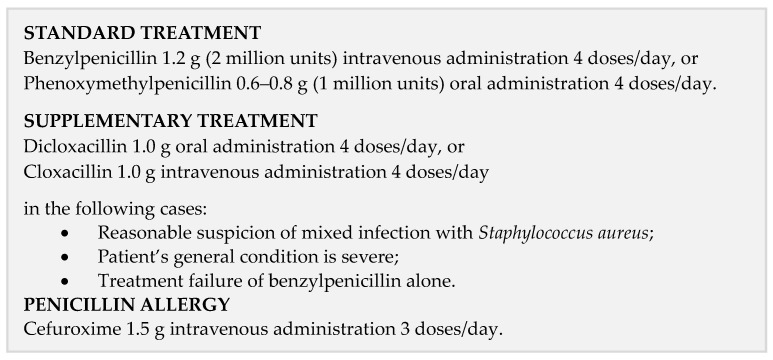
Regional antibiotic guidelines.

**Table 1 antibiotics-13-01021-t001:** Characteristics of the 777 patients admitted to the ED diagnosed with cellulitis and other infections.

Characteristic	Patients with Cellulitis	Missing Data	Patients with Other Infections	Missing Data	OR (95% CI)	*p*-Value
Total, n (%)	77 (9.9)	0	700 (90.1)	0		
Demographics						
Age group, n (%) 18–40 41–60 61–80 ≥81	6 (7.8) 23 (29.9) 34 (44.2) 14 (18.2)	0	57 (8.1) 108 (15.4) 337 (48.1) 198 (28.3)	0	0.95 (0.40–2.29) **2.33 (1.38–3.96)** 0.85 (0.53–1.37) 0.56 (0.31–1.03)	0.915 **0.002** 0.507 0.062
Sex, male, n (%)	53 (58.8)	0	378 (54.0)	0	**1.88 (1.14–3.12)**	**0.014**
Vital parameters						
Fever *, n (%)	15 (20.0)	2	231 (33.1)	2	**0.51 (0.28–0.91)**	**0.023**
Oxygen saturation ≤ 96, n (%)	34 (44.7)	1	406 (58.2)	2	**0.58 (0.36–0.94)**	**0.026**
Respiration rate ≥ 22/min, n (%)	10 (13.5)	3	216 (31.0)	2	**0.35 (0.18–0.69)**	**0.003**
Heart rate > 90 beats/min, n (%)	35 (46.1)	1	353 (50.4)	0	0.84 (0.52–1.35)	0.469
Systolic blood pressure ≤ 100, n (%)	5 (6.6)	1	51 (7.3)	2	0.89 (0.35–2.31)	0.816
qSOFA score ≥ 2, n (%)	0 (0.0)	3	34 (4.9)	6	N/A	N/A
Lifestyle						
Alcohol overuse **, n (%)	9 (11.7)	0	77 (11.5)	31	1.02 (0.49–2.12)	0.963
Current smoker, n (%)	7 (9.1)	0	128 (19.1)	28	**0.43 (0.19–0.95)**	**0.036**
Body mass index > 30, n (%)	27 (45.0)	17	122 (23.6)	182	**2.66 (1.54–4.59)**	**<0.001**
Underlying disease						
Type 2 diabetes, n (%)	21 (27.3)	0	110 (15.7)	0	**2.01 (1.17–3.46)**	**0.011**
Cardiovascular disease, n (%)	37 (48.1)	0	285 (40.7)	0	1.35 (0.84–2.16)	0.216
Prior cellulitis						
One prior case of cellulitis, n (%)	11 (14.3)	0	45 (6.8)	36	**2.29 (1.13–4.65)**	**0.021**
>1 prior cases of cellulitis, n (%)	26 (33.8)	0	22 (3.3)	36	**14.88 (7.88–28.08)**	**<0.001**
Blood samples						
Elevated leukocytes ***, n (%)	58 (75.3)	0	526 (75.1)	0	1.01 (0.59–1.74)	0.972
Elevated neutrophils ***, n (%)	33 (43.4)	1	373 (53.8)	7	0.66 (0.41–1.06)	0.086
C-reactive protein, n (%) Low ≤20 mg/L Moderate 21–99 mg/L High ≥100 mg/L	11 (14.3) 19 (24.7) 47 (61.0)	0	92 (13.1) 216 (30.9) 392 (56.0)	0	1.10 (0.56–2.16) 0.73 (0.43–1.26) 1.23 (0.76–1.99)	0.779 0.265 0.398

* fever = temperature ≥ 38 °C. ** above the recommended weekly amount from the Danish Government (2019): 1–7 doses maximum for women and 8–14 doses maximum for men. One dose = 12 g (1.5 cL) of alcohol. *** values > 8.8 × 10^9^/L. OR: odds ratio; CI: confidence interval. Bold values indicate significant difference between groups (*p* < 0.05).

**Table 2 antibiotics-13-01021-t002:** Antibiotic treatment of the 77 patients with cellulitis admitted to the ED.

	Within 30 Days Prior to Admission	4 h After Admission	48 h After Admission	5 Days After Admission
Received antibiotic treatment, n (%)	42 (54.5)	74 (96.1)	76 (98.7)	76 (98.7)
Peroral antibiotic treatment, n (%)	42 (100)	16 (21.6)	37 (48.7)	61 (80.3)
Intravenous antibiotic treatment, n (%)	2 (4.8)	61 (82.4)	40 (52.6)	16 (21.1)
Beta-lactamase-sensitive penicillins *, n (%)	24 (57.1)	61 (82.4)	65 (85.5)	61 (80.3)
Beta-lactamase-resistant penicillins **, n (%)	21 (50.0)	33 (44.6)	38 (50.0)	41 (53.9)
Other ***, n (%)	8 (19.0)	12 (16.2)	8 (10.5)	9 (11.8)

* ATC J01CE (phenoxymethylpenicillin and benzylpenicillin). ** ATC J01CF (dicloxacillin and cloxacillin). *** ATC J01CR05 (piperacillin/tazobactam); ATC J01DB (cephalosporins); ATC J01CA (pivampicillin and ampicillin); ATC J01X (metronidazole); ATC J01F (lincosamides and macrolides); ATC J01E (sulfonamides); ATC J01G (aminoglycosides).

**Table 3 antibiotics-13-01021-t003:** Prognosis for the 777 patients admitted to the ED diagnosed with an infection.

	Patients with Cellulitis	Patients with Other Infections	OR (95% CI)	*p*-Value
Total, n (%)	77 (9.9)	700 (90.1)		
Discharged to home within 48 h	38 (49.4)	231 (33.0)	**1.98 (1.23–3.18)**	**0.005**
Discharged from the ED to other units after 48 h Medical ward, n (%) Surgical ward, ICU and other, n (%)	35 (89.7) 4 (10.3)	407 (86.8) 62 (13.2)	1.33 (0.46–3.90) 0.75 (0.26–2.18)	0.598 0.598
Readmission within 30 days after discharge, n (%)	20 (26.0)	149 (21.4)	1.29 (0.75–2.22)	0.352
30-day mortality, n (%)	3 (3.9)	62 (8.9)	0.42 (0.13–1.36)	0.148
			**IRR (95% CI)**	
Length of hospital stay in days, median (IQR)	2 (1–4)	3 (1–6)	0.82 (0.62–1.07)	0.149

ED: Emergency Department; ICU: intensive care unit; IQR: interquartile range; OR: odds ratio; CI: confidence interval; IRR: incidence rate ratio. Bold values indicate significant difference between groups (*p* < 0.05).

## Data Availability

Due to Danish laws on personal data, data cannot be shared publicly. To request data, please contact the corresponding author for more information. The person responsible for the research was the principal investigator (HS-A) and corresponding author (AKW) in collaboration with the University Hospital of Southern Denmark. This organization owns the data and can provide access to the final dataset.

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
