# Peer review of "Characteristics and Antibiotic Treatment of Patients with Cellulitis in the Emergency Department"

_antibiotics, 2024, doi:10.3390/antibiotics13111021_

Round 1
Reviewer 1 Report
Comments and Suggestions for Authors
Interesting study on antibiotics used in the treatment of erysipelas. However, there is a clear lack of data regarding microbiological results. It is a certain fact that the study is prospective and from the outset microbiological samples were not taken into account. However, it appears in the discussion that to argue the therapeutic decisions, the authors base themselves on the results of the literature on the microbiology of erysipelas. It should be added a clarification in the material and methods paragraph that in this study microbiological results were not considered.
On the other hand, many studies consider that no laboratory workup is required for the diagnosis of erysipelas. Leukocytosis, elevated ESR and C-reactive protein are common but will not change the management or the treatment plan for most otherwise healthy individuals. Blood cultures have a low yield and are not routinely obtained; however, consider blood work and culture in the immunocompromised, ill-appearing patient. Also, consider extensive workup in patients who may be intravenous drug abusers, patients with prosthetic heart valves, and those with other intravascular devices. Patients who are septic will require full workup and resuscitation. (Michael Y, Shaukat NM. Erysipelas. [Updated 2023 Aug 7]. In: StatPearls [Internet]. Treasure Island (FL): StatPearls Publishing; 2024 Jan-. Available from: https://www.ncbi.nlm.nih.gov/books/NBK532247/)
Author Response
Dear reviewer,
Thank you very much for taking the time to provide feedback on our manuscript. We appreciate the effort you have put in to reviewing our work. Your comments were very valuable, and helped us improve the manuscript.
We have carefully addressed your suggestions below:
- Interesting study on antibiotics used in the treatment of erysipelas. However, there is a clear lack of data regarding microbiological results. It is a certain fact that the study is prospective and from the outset microbiological samples were not taken into account. However, it appears in the discussion that to argue the therapeutic decisions, the authors base themselves on the results of the literature on the microbiology of erysipelas. It should be added a clarification in the material and methods paragraph that in this study microbiological results were not considered.
Thank you very much for your comment on the lack of data regarding microbiological results. We agree that the manuscript was missing a statement acknowledging that no microbiological samples were considered, and have addressed this issue in our manuscript, see line 118-120.
Once again, we appreciate your review and believe that your feedback has improved our manuscript.
Kind regards,
Aaron K. Wiederhold
Corresponding author
Reviewer 2 Report
Comments and Suggestions for Authors
This study is generally well-written, and the results are presented clearly and concisely. Please find my specific comments below.
1. The background of the study could be more detailed to clarify the significance and necessity of this study, identifying the research gap it aims to fill.
2. The proportion of missing data for some variables is higher than 5% (e.g., lifestyle), which should be addressed rather than directly analyzed.
Author Response
Dear reviewer,
Thank you for taking the time to review our manuscript. We greatly appreciate the effort you invested in providing feedback. Your insights were very helpful and contributed significantly to enhancing our work.
We have carefully reviewed your comments and made following changes to our manuscript:
- The background of the study could be more detailed to clarify the significance and necessity of this study, identifying the research gap it aims to fill.
Thank you very much for your comment. We agree that additional background information clarifying the significance of our study would strengthen the introduction section. In response to your comment, we added further explanation regarding the importance of additional research on the characteristics and treatment of patients with cellulitis in the introduction section, see line 54-57.
- The proportion of missing data for some variables is higher than 5% (e.g., lifestyle), which should be addressed rather than directly analyzed.
We highly value your thoroughness. We agree on your comment and have addressed the issue of the large proportion of missing data in our discussion section, see line 279-282.
Once again, we appreciate your review and believe that your feedback has improved our manuscript.
Kind regards,
Aaron K. Wiederhold
Corresponding author